# Finite Groups for the Kummer Surface: The Genetic Code and a Quantum Gravity Analogy

**Michel Planat** [1,*], **David Chester** [2], **Raymond Aschheim** [2], **Marcelo M. Amaral** [2], **Fang Fang** [2] and **Klee Irwin** [2]

1   Institut FEMTO-ST CNRS UMR 6174, Université de Bourgogne/Franche-Comté, 15 B Avenue des Montboucons, F-25044 Besançon, France
2   Quantum Gravity Research, Los Angeles, CA 90290, USA; DavidC@QuantumGravityResearch.org (D.C.); raymond@QuantumGravityResearch.org (R.A.); Marcelo@quantumgravityresearch.org (M.M.A.); fang@quantumgravityresearch.org (F.F.); Klee@quantumgravityresearch.org (K.I.)
*   Correspondence: michel.planat@femto-st.fr

**Abstract:** The Kummer surface was constructed in 1864. It corresponds to the desingularization of the quotient of a 4-torus by 16 complex double points. Kummer surface is known to play a role in some models of quantum gravity. Following our recent model of the DNA genetic code based on the irreducible characters of the finite group $G_5 := (240, 105) \cong \mathbb{Z}_5 \rtimes 2O$ (with $2O$ the binary octahedral group), we now find that groups $G_6 := (288, 69) \cong \mathbb{Z}_6 \rtimes 2O$ and $G_7 := (336, 118) \cong \mathbb{Z}_7 \rtimes 2O$ can be used as models of the symmetries in hexamer and heptamer proteins playing a vital role for some biological functions. Groups $G_6$ and $G_7$ are found to involve the Kummer surface in the structure of their character table. An analogy between quantum gravity and DNA/RNA packings is suggested.

**Keywords:** kummer surface; DNA genetic code; hexamers and pentamers; informationally complete characters; finite groups; hyperelliptic curve

## 1. Introduction

In a recent paper we found that the 22 irreducible characters of the group $G_5 := (240, 105) \cong \mathbb{Z}_5 \rtimes 2O$, with $2O$ the binary octahedral group, could be made in one-to-one correspondence with the DNA multiplets encoding the proteinogenic amino acids [1]. The cyclic group $\mathbb{Z}_5$ features the five-fold symmetry of the constituent bases, see Figure 1a. An important aspect of this approach is that the irreducible characters of $G_5$ may be seen as 'magic' quantum states carrying minimal and complete quantum information, see [1–3] for the meaning of these concepts. It was also shown that the physical structure of DNA was reflected in some of the entries of the character table including the Golden ratio, the irrational number $\sqrt{2}$, as well as the four roots of a quartic polynomial.

In molecular biology, there exists a ubiquitous family of RNA-binding proteins called LSM proteins whose function is to serve as scaffolds for RNA oligonucleotides, assisting the RNA to maintain the proper three-dimensional structure. Such proteins organize as rings of six or seven subunits. The Hfq protein complex was discovered in 1968 as an Escherichia coli host factor that was essential for replication of the bacteriophage $Q\beta$ [4], it displays an hexameric ring shape shown in Figure 1b.

It is known that in the process of transcription of DNA to proteins through messenger RNA sequences (mRNAs), there is an important step performed in the spliceosome [5]. It consists of removing the non-coding intron sequences for obtaining the exons that code for the proteinogenic amino acids. A ribonucleoprotein (RNP)—a complex of ribonucleic acid and RNA-binding protein—plays a vital role in several biological functions that include transcription, translation, the regulation of gene expression and the metabolism of

RNA. Individual LSm proteins assemble into a six- or seven-member doughnut ring which usually binds to a small RNA molecule to form a ribonucleoprotein complex.

Thus, while five-fold symmetry is inherent to the bases $A$, $T$, $G$, $C$—the building blocks of DNA—six-fold and seven-fold symmetries turn out to be the rule at the level of the spliceosome [6]. Of the five small ribonucleoproteins, four of them called U1, U2, U4 and U5, contain an heptamer ring, whereas the U6 contains a specific Lsm2-8 heptamer with seven-fold symmetry. A specific Lsm heptameric complex Lsm1-7 playing a role in mRNA decapping is shown in Figure 1c [7].

Observe that six-fold rings are also present in other biological functions such as genomic DNA replication [8,9]. The minichromosome maintenance complex (MCM) hexameric complex (Mcm2–7) forms the core of the eukaryotic replicative helicase. Eukaryotic MCM consists of six gene products, Mcm2-7, which form a heterohexamer [9]. Deregulation of MCM function has been linked to genomic instability and a variety of carcinomas.

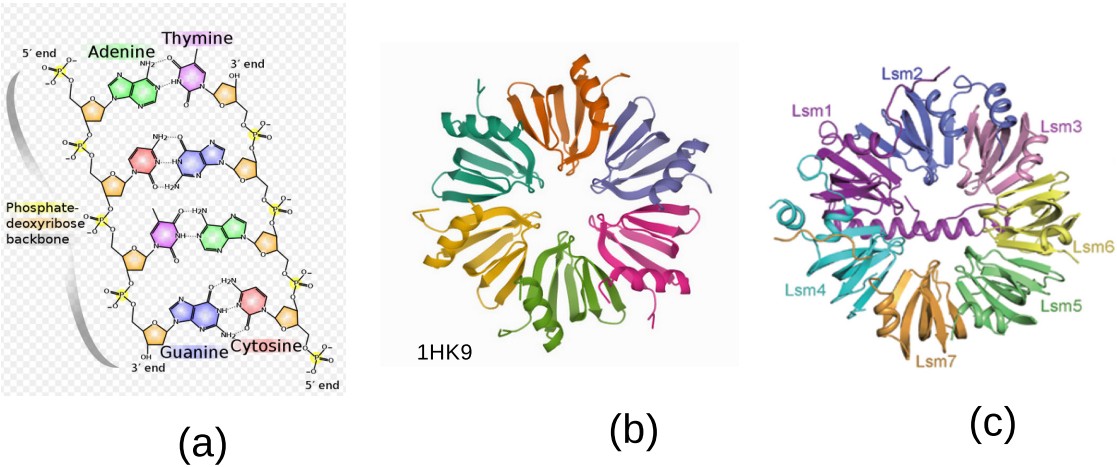

**Figure 1.** (**a**) Five-fold symmetry in the DNA, (**b**) six-fold symmetry in the LSM protein complex Hfq [4], (**c**) seven-fold symmetry of the Lsm1-7 complex in the spliceosome [7].

In this paper, in order to approach these biological issues—the hexamer and pentamer rings—we generalize our previous model of the DNA/RNA, which has been based on the five-fold symmetry group $G_5$, to models of DNA/RNA complexes, based on the six-fold symmetry group $G_6 := (288, 69) \cong \mathbb{Z}_6 \times 2O$ and the seven-fold symmetry group $G_7 := (336, 118) \cong \mathbb{Z}_7 \times 2O$.

What corresponds to the quartic curve, derived from some of the entries in the character table of $G_5$, is a genus 2 hyperelliptic curve derived from the character table of $G_6$ or $G_7$, underlying the so-called Kummer surface, a gem of algebraic geometry [10,11]. The Kummer surface is a prototypic example of a $K_3$ surface—a Calabi–Yau manifold of complex dimension two—and as such it is part of models in string theory and/or quantum gravity [12,13].

Section 2.1 is a brief introduction to elliptic and hyperelliptic curves defined over any field. In Sections 2.2 and 2.3, the main objective is to describe a construction of the Kummer surface $\mathcal{K}$ based on the character table of the groups $G_6$ or $G_7$. One identifies the 16 double points of $\mathcal{K}$ as the 16 two-torsion points of a genus two hyperelliptic curve $\mathcal{C}$ and one provides an explicit description of the group law and of the Kummer surface.

In Section 3, a new encoding of the proteinogenic amino acids by the irreducible characters and the corresponding representations of the group $G_7$ is described. It improves the description obtained in [1] from the 22 irreducible characters of group $G_5$.

In Section 4, we browse over some applications of the Kummer surface to models of quantum gravity.

## 2. The Hyperelliptic Curve and the Attached KUMMER Surface from Groups $G_6$ and $G_7$

Let us first recall an important aspect of our previous work. Let $G$ be a finite group with $d$ conjugacy classes. An irreducible character $\kappa = \kappa_r$ corresponding to a $r$-dimensional representation of $G$ carries quantum information [1–3]. It may be calculated thanks to the action of elements of a $d$-dimensional Pauli group $\mathcal{P}_d$ acting on $\kappa$. In other words, the character $\kappa$ may be seen as the 'magic state' of a quantum computation [2,14].

In a concrete way, one defines $d^2$ one-dimensional projectors $\Pi_i = |\psi_i\rangle\langle\psi_i|$, where the $|\psi_i\rangle$ are the $d^2$ states obtained from the action of $\mathcal{P}_d$ on $\kappa$, and one calculates the rank of the Gram matrix $\mathcal{G}$ with elements $\mathrm{tr}(\Pi_i\Pi_j)$. A Gram matrix $\mathcal{G}$ with rank equal to $d^2$ is the signature of a minimal informationally complete quantum measurement (or MIC), see e.g., [Section 3] of [1] for more details.

As in our previous work, in a character table, we will display the measure of quantum information of a character $\kappa_r = \kappa$ as the rank of the attached Gram matrix $\mathcal{G}$.

### 2.1. Excerpts about Elliptic and Hyperelliptic Curves

Let us consider a curve $\mathcal{C}$ defined with the algebraic equation $y^2 + h(x)y = f(x)$ where $h(x)$ and $f(x)$ are finite degree polynomials with elements in a field $K$.

For an elliptic curve—let us use the notation $\mathcal{E}$ instead of $\mathcal{C}$ for this case—polynomials $h(x)$ and $f(x)$ are of degrees 1 and 2, respectively and the genus of $\mathcal{E}$ is $g = 1$. In the Weierstrass form of an elliptic curve, one takes $h(x) = a_1 x + a_3$ and $f(x) = x^3 + a_2 x^2 + a_4 x + a_6$ so that $\mathcal{E}$ is specified with the sequence $[a_1, a_2, a_3, a_4, a_5]$ of elements of $K$. There is a rich literature about elliptic curves defined over rational fields $\mathbb{Q}$, complex fields $\mathbb{C}$, number fields or general fields. Explicit results are documented in tables such as the Cremona table [15] or can be obtained from a mathematical software such as Magma [16].

An elliptic curve (as well as a hyperelliptic curve) may be viewed as embedded in a weighted projective space, with weights 1, $g + 1$ and 1, in which the points at infinity are non-singular. In the present work, one meets genus 2 curves for which there exists a set of 16 double points leading to the construction of a Kummer surface. All curves of genus 2 are hyperelliptic but generic curves of genus $g > 2$ are not. Again references [15,16] are basic references for explicit results.

There are plenty known invariants of a (hyper)elliptic curve $\mathcal{C}$ over a field. One of them is the conductor $N$ of an elliptic curve $\mathcal{E}$ seen as an abelian variety $A$. For $A$ defined over $\mathbb{Q}$, the conductor is the positive integer whose prime factors are the primes where $A$ has a bad reduction. The conductor characterizes the isogeny class of $A$ so that the curves $\mathcal{E}$ over $\mathbb{Q}$ may be classified according to the isogeny classes. Another important invariant is the Mordell-Weil group of $A$ which is the group $A(\mathbb{Q})$ of $\mathbb{Q}$-rational points of $\mathcal{E}$. Weil proved that $A(\mathbb{Q})$ is finitely generated with a unique decomposition of the form $A(\mathbb{Q}) \cong \mathbb{Z}^{r_A} \oplus T$, where the finite group $T$ is the torsion subgroup and $r_A$ is the rank of $A$. This result was later generalized to the elliptic curves defined over any field $K$. For a hyperelliptic curve, the invariant in the Weierstrass equation of $\mathcal{C}$ called the discriminant $\Delta$ may be defined over any field $K$. For an elliptic curve, this leads to the $j$-invariant $j = c_4^3/\Delta$ with $c_4$ a polynomial function of the coefficients in the Weierstrass form.

The main applications of elliptic curves are in the field of public-key cryptography. For hyperelliptic curves, one makes use of the Jacobian as the abelian group in which to do arithmetic, as one uses the group of points on an elliptic curve.

From now we describe how genus 1 curves (elliptic curves) and genus 2 curves (hyperelliptic curves of the Kummer type) arise in the character table of groups of signature $G_i \cong \mathbb{Z}_i \rtimes 2O$, $i = 5$, 6 and 7. Other finite groups and curves of genus $g > 2$ built from the character table of a finite group $G$ are worthwhile to be investigated in the future.

Let us illustrate our description with the genus 1 hyperelliptic curve introduced in the context of our model of the genetic code based on the group $G_5 := (240, 105)$ in which $h(x) = 0$ and $f(x) = x^4 - x^3 - 4x^2 + 4x + 1$ [Section 5] of [1]. Seeing this curve over the rationals, one learns from Magma [16] that the conductor is $N = 300$ and the discriminant is $\Delta = 18000$. A look at the Cremona table for elliptic curves [15] allows

us to put our curve in the isogeny class of Cremona reference '300d1'. The Weierstass form is $y^2 = x^3 - x^2 - 13x + 22$ and the Mordell-Weil group is the group of infinite cardinality $\mathbb{Z} \times \mathbb{Z}_2$.

Now we use this knowledge to investigate some properties of the character tables of group $G_6$ and $G_7$.

### 2.2. The Group $G_6 := (288, 69) \cong \mathbb{Z}_6 \rtimes 2O$

One first considers the group $G_6 := (288, 69) \cong \mathbb{Z}_6 \rtimes 2O$, with $2O$ the binary octahedral group. The structure of the character table is shown in Table 1. All characters are neither faithful nor informationally complete since the rank of the Gram matrix is never $d^2 = 30^2$ for any character.

**Table 1.** For the group $G_6 := (288, 69) \cong \mathbb{Z}_6 \rtimes 2O$, the table provides the dimension of the representation, the rank of the Gram matrix obtained under the action of the 30 -dimensional Pauli group and the entries involved in the characters. All characters are neither faithful nor informationally complete. The notation is $I = \exp(2i\pi/4)$, $z_1 = -\sqrt{2}$, $z_2 = I\sqrt{2}$ and $z_3 = -2\cos(\pi/9)$.

| (288,69) | dimension | 1 | 1 | 1 | 1 | 2 | 2 | 2 | 2 | 2 | 2 |
|---|---|---|---|---|---|---|---|---|---|---|---|
| $\mathbb{Z}_6 \rtimes (\mathbb{Z}_2 \cdot S_4)$ | d-dit, d = 30 | 31 | 796 | 867 | 867 | 882 | 882 | 880 | 897 | 897 | 880 |
| | char | Cte | Cte | I | I | Cte | Cte | $z_1$ | $z_2$ | $z_2$ | $z_1$ |
| (288,69) | dimension | 2 | 2 | 2 | 2 | 2 | 2 | 4 | 4 | 4 | 4 |
| | d-dit, d = 30 | 885 | 885 | 885 | 885 | 885 | 885 | 876 | 878 | 899 | 899 |
| | char | $z_3$ | $z_3$ | $z_3$ | $z_3$ | $z_3$ | $z_3$ | Cte | Cte | I | I |
| (288,69) | dimension | 4 | 4 | 4 | 4 | 4 | 4 | 6 | 6 | 6 | 6 |
| | d-dit, d = 30 | 877 | 878 | 885 | 885 | 885 | 885 | 885 | 885 | 880 | 880 |
| | char | Cte | Cte | $z_3$ | $z_3$ | $z_3$ | $z_3$ | $z_3$ | $z_3$ | Cte | Cte |

Some characters contain entries with complex numbers $I$ or $z_2 = I\sqrt{2}$. There are 12 characters containing entries with $z_3 = -2\cos(\pi/9)$ featuring the angle $\pi/9$. We now show an important characteristic of such characters. As an example, let us write the character number 11 as obtained from Magma [16]

$$\kappa_{11} = [2, 2, -2, -2, -1, 2, -2, 0, 0, 1, -1, 1, 0, 0, 0, 0, z_3, z_3\#2, z_3\#4, -1,$$
$$1, -z_3\#2, -z_3\#4, -z_3\#4, z_3\#2, z_3, -z_3, z_3\#4, -z_3, -z_3\#2]$$

where # denotes the algebraic conjugation, i.e., #$k$ indicates replacing the root of unity $w$ by $w^k$. The non-constant (but real) entries are $k_{\pm l} = \pm z_3 \# l$, with $l = 1, 2$ or 4. We obtain $k_1 = -2\cos(\pi/9)$ and $k_{2,4} = \cos(\pi/9) \mp \cos(2\pi/9) \pm \cos(4\pi/9)$ and $k_1 + k_2 + k_4 = 0$.

Following our approach in [1], we construct an hyperelliptic curve $\mathcal{C}_6$ of the form $y^2 = \prod_{\pm l}(x - k_l) = f(x)$. In an explicit way, it is

$$\mathcal{C}_6 : \quad y^2 = x^6 - 6x^4 + 9x^2 - 1, \tag{1}$$

a genus 2 hyperelliptic curve. Using Magma [16], one gets the polynomial definition of the Kummer surface $S(x_1, x_2, x_3, x_4)$ as

$$S(x_1, x_2, x_3, x_4) = 36x_1^4 + 4x_1^3 x_4 - 24x_1^2 x_2^2 + 220x_1^2 x_3^2 - 36x_1^2 x_3 x_4 - 8x_1 x_2^2 x_3$$
$$+ 24x_1 x_3^2 x_4 - 4x_1 x_3 x_4^2 + 4x_2^4 - 36x_2^2 x_3^2 + x_2^2 x_4^2 + 24x_3^4 - 4x_3^3 x_4.$$

The desingularization of the Kummer surface is obtained in a simple way by restricting the product $f(x) = \prod_{\pm l}(x - k_l)$ to the five first factors with indices $\pm 1$, $\pm 2$ and 4.

One embeds $\mathcal{C}_6$ in a weighted projective plane, with weights 1, $g + 1$, and 1, respectively on coordinates $x$, $y$ and $z$. Therefore, point triples are such that $(x : y : z) = (\mu x : \mu y : \mu z)$, $\mu$ in the field of definition, and the points at infinity take the form $(1 : y : 0)$. Below, the software Magma is used for the calculation of points of $\mathcal{C}_6$ [16]. For the points of $\mathcal{C}_6$, there

is a parameter called 'bound' that loosely follows the heights of the $x$-coordinates found by the search algorithm.

It is found that the corresponding Jacobian of $\mathcal{C}_6$ has $16 = 6 + 10$ points as follows
* the 6 points bounded by the modulus 1:

$Id := (1, 0, 0)$, $K_{\pm 1} := (x - k_{\pm 1}, 0, 1)$, $K_{\pm 2} := (x - k_{\pm 2}, 0, 1)$ and $K_4 = (k_4, 0, 1)$.
* the 10 points of modulus $> 1$:

$a_1 := K_{-1} + K_4$, $a_2 := K_1 + K_{-1}$, $a_3 := K_1 + K_{-2}$, $a_4 := K_1 + K_{-1} + K_2$, $a_5 := K_1 + K_{-1} + K_{-2}$, $a_6 := K_1 + K_4$, $a_7 := K_{-1} + K_2$, $a_8 := K_1 + K_{-1} + K_4$, $a_9 := K_{-1} + K_{-2}$ and $a_{10} := K_1 + K_{-1} + K_2$,

The 16 points organize as a commutative group isomorphic to the maximally abelian group $\mathbb{Z}_2^4$ as shown in the following Jacobian addition table 2.

**Table 2.** The structure of the addition table for the 16 singular Jacobian points of the hyperelliptic curves $\mathcal{C}_6$ and $\mathcal{C}_7$.

| A | B | C | D |
|---|---|---|---|
| B | A | D | C |
| C | D | A | B |
| D | C | B | A |

where the blocks are given explicitly as

$$
A : \begin{bmatrix} Id & K_1 & K_{-1} & a_2 \\ K_1 & Id & a_2 & K_{-1} \\ K_{-1} & a_2 & Id & K_1 \\ a_2 & K_{-1} & K_1 & Id \end{bmatrix}, \quad
B : \begin{bmatrix} K_2 & a_{10} & a_7 & a_4 \\ a_{10} & K_2 & a_4 & a_7 \\ a_7 & a_4 & K_2 & a_{10} \\ a_4 & a_7 & a_{10} & K_2 \end{bmatrix},
$$

$$
C : \begin{bmatrix} K_{-2} & a_3 & a_9 & a_5 \\ a_3 & K_{-2} & a_5 & a_3 \\ a_9 & a_5 & K_{-2} & a_3 \\ a_5 & a_9 & a_3 & K_{-2} \end{bmatrix}, \quad
D : \begin{bmatrix} a_8 & a_1 & a_6 & K_4 \\ a_1 & a_8 & K_4 & a_6 \\ a_6 & K_4 & a_8 & a_1 \\ K_4 & a_6 & a_1 & a_8 \end{bmatrix}.
$$

As a whole, one can check that there are only 48 points in the Jacobian $J(\mathcal{C}_6)$. They organize at the group $\mathbb{Z}_3 \times \mathbb{Z}_2^3$, i.e., three copies of the group of singular points.

*2.3. The Group $G_7 := (336, 118) \cong \mathbb{Z}_7 \times 2O$*

Let us consider the group $G_7 := (336, 118) \cong \mathbb{Z}_7 \rtimes 2O$, with $2O$ the binary octahedral group. The structure of the character table is shown in the table of Section 3 about a refined model of the genetic code. Except for the singlets, the irreducible characters of $G_7$ are informationally complete (with rank of the Gram matrix equal to $d^2 = 29^2$ for any character). Only the first two singlets are exceptions. The entries involved in the characters are $z_1 = 2\cos(2\pi/7)$, $z_2 = 2z_1$, $z_3 = -6\cos(\pi/7)$, $z_4 = \sqrt{2}$ and $z_5 = 2\cos(2\pi/21)$ featuring the angles $2\pi/8$ (in $z_4$), $2\pi/7$ and $2\pi/21$. There are 9 faithful characters over the 10 quartets.

A summary of the elliptic and genus 2 hyperelliptic curves that can be defined from $G_7$ is in Table 3.

**Table 3.** The algebra for the character table of group $G_7 := (336, 118)$. In column 1 are the characters in question. Column 2 provides the powers of the entries $z_i$, $i = 1, 2, 3$ or 5. The $z_i$ are $z_1 = 2\cos(2\pi/7)$, $z_2 = 2z_1$, $z_3 = -6\cos(\pi/7)$, $z_4 = \sqrt{2}$ and $z_5 = 2\cos(2\pi/21)$. Column 3 explains the polynomial $f(x)$ whose roots are the powers of a selected $z_i$. When $f(x)$ is an elliptic curve defined over the rationals the Cremona reference is in column 4. If $f(x)$ is a sextic polynomial it leads to a Kummer surface.

| Character | $z_i$ Powers | $f(x)$ Polynomial | Cremona Ref. |
|:---:|:---:|:---:|:---:|
| 4–6 | $z_1 : [1, 2, 3]$ | $x^3 + x^2 - 2x - 1$ | $784i_1$ |
| 18–20 | $z_1 : [1, 2, 3]$ | . | . |
| . | $z_2 : [1, 2, 3]$ | $x^3 + 2x^2 - 8x - 8$ | $3136x_1$ |
| . | $z_{1,2}$ | $x^6 + 3x^5 - 8x^4 - 21x^3 + 6x^2 + 24x + 8$ | Kummer |
| 27–29 | $z_1 : [1, 2, 3]$ | . | . |
| . | $z_3 : [1, 2, 3]$ | $x^3 + 3x^2 - 18x - 27$ | $1764j_1$ |
| . | $z_{1,3} : [1, 2, 3]$ | $x^6 + 4x^5 - 17x^4 - 52x^3 + 6x^2 + 72x + 27$ | Kummer |
| 9–14 & 21–26 | $z_{1,2}$ | . | . |
| | $z_5 : [1, 2, 4, 5, 8, 10]$ | $x^6 - x^5 - 6x^4 + 6x^3 + 8x^2 - 8x + 1$ | Kummer |

For instance, characters 4 to 6 as obtained from Magma [16] contains non-constant entries with $z_1$, $z_1$#2 and $z_1$#3. With the polynomial $f(x) = (x - z_1)(x - z_1\#2)(x - z_1\#3)$ one defines the elliptic curve $y^2 = f(x)$ over the rationals whose conductor $N$ and discriminant $\Delta$ are equal to 784 and whose $j$-invariant equals 1792. It corresponds to the isogeny class of the curve $784i_1$ in the Cremona table [15].

There are 12 characters containing entries with $z_5$. We now show an important characteristic of such characters. As an example, let us write the character number 9 as obtained from Magma [16]

$$\kappa_9 = [2, 2, -1, 2, 0, -1, z_1\#3, z_1\#2, z_1, 0, 0, z_1\#2, z_1\#3, z_1, z_5, z_5\#4, z_5\#8,$$
$$z_5\#10, z_5\#2, z_5\#5, z_1\#2, z_1, z_1\#3, z_5\#8, z_5\#5, z_5\#2, z_5, z_5\#4, z_5\#10]$$

The hyperelliptic curve $\mathcal{C}_7 : y^2 = f(x)$ attached to the Kummer surface defined over the group $G_7 := (336, 118)$ is

$$y^2 = f(x) = (x - k)(x - l)(x - m)(x - n)(y - o)(y - p), \qquad (2)$$

with $k = 2\cos(10\pi/21)$, $l = 2\cos(4\pi/21)$, $m = 2\cos(16\pi/21)$, $n = 2\cos(2\pi/21)$ and $o = -2\cos(\pi/21)$ (as above) and $p = \cos(\pi/21) + \cos(8\pi/21) - \cos(6\pi/21)$. The sum of roots of the sextic curve $f(x)$ equals 1.

The defining polynomial can be given an explicit expression over the rational field

$$f(x) = x^6 - x^5 - 6x^4 + 6x^3 + 8x^2 - 8x + 1,$$

leading to the Kummer surface

$$K3(x_1, x_2, x_3, x_4) = 32x_1^4 - 24x_1^3 x_2 + 96x_1^3 x_3 - 4x_1^3 x_4 + 24x_1^2 x_2^2$$
$$- 196x_1^2 x_2 x_3 + 16x_1^2 x_2 x_4 + 240x_1^2 x_3^2 - 32x_1^2 x_3 x_4 + 4x_1 x_2^3 - 24x_1 x_2^2 x_3$$
$$- 12x_1 x_2 x_3 x_4 + 12x_1 x_3^3 + 24x_1 x_3^2 x4 - 4x_1 x_3 x_4^2 - 4x_2^4 + 32x_2^3 x_3 - 32x_2^2 x_3^2$$
$$+ x_2^2 x_4^2 - 24x_2 x_3^3 + 2x_2 x_3^2 x_4 + 25x_3^4 - 4x_3^3 x_4.$$

A section at constant $x_4$ of this Kummer surface is given in Figure 2b using the MathMod software [17].

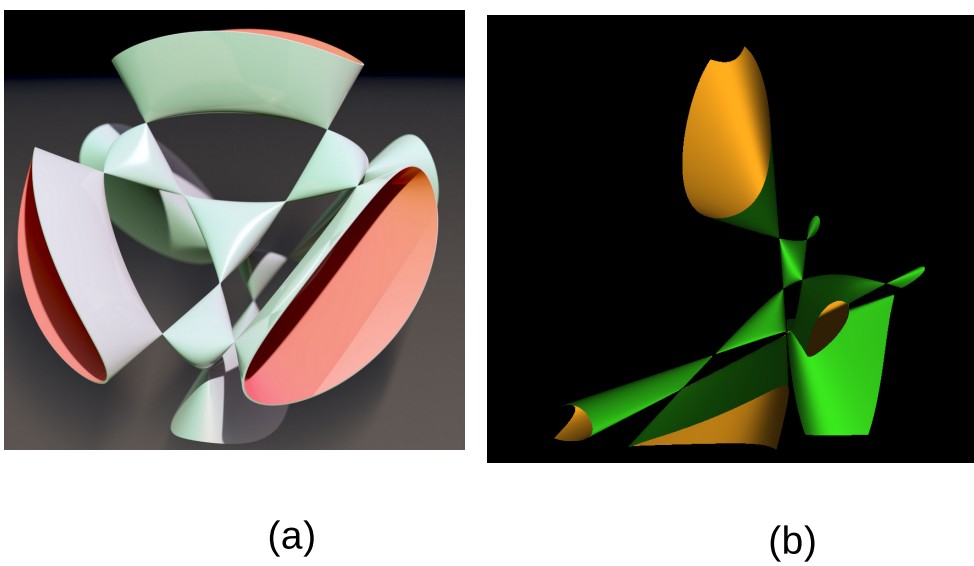

(a)                                                          (b)

**Figure 2.** (**a**) A standard plot of the Kummer surface in its 3-dimensional projection, (**b**) a section at constant $x_4$ of the Kummer surface defined in Section 2.3.

The desingularization of the Kummer surface is obtained by restricting the product in the polynomial $f(x)$ to the five first factors. It is found that the corresponding Jacobian of $C_7$ has $16 = 6 + 10$ points as follows

\* the 6 points bounded by the modulus 1:

$Id := (1,0,0)$, $K := (x-k,0,1)$, $L := (x-l,0,1)$, $N := (x-n,0,1)$, $M := (x-m,0,1)$ and $O := (x-o,0,1)$.

\* the 10 points of modulus $> 1$: $a_1 := K+L$, $a_2 := K+M$, $a_3 := K+N$, $a_4 := K+O$, $a_5 := L+N$, $a_6 := K+L+M$, $a_7 := K+L+N$, $a_8 := K+L+O$, $a_9 := 2K+L+M$ and $a_{10} := 2K+L+O$.

More explicitly, for instance, $K+L = (x^2 - (k+l)x + 2\cos(2\pi/7), 0, 2)$.

The 16 points organize as a commutative group isomorphic to the maximally abelian group $\mathbb{Z}_2^4$ as shown in Table 2 with the entries

$$A : \begin{bmatrix} Id & K & L & a_1 \\ K & Id & a_1 & L \\ L & a_1 & Id & K \\ a_1 & L & K & Id \end{bmatrix}, \quad B : \begin{bmatrix} N & a_3 & a_5 & a_7 \\ a_3 & N & a_7 & a_5 \\ a_5 & a_7 & N & a_3 \\ a_7 & a_5 & a_3 & N \end{bmatrix}$$

$$C : \begin{bmatrix} M & a_2 & a_9 & a_6 \\ a_2 & M & a_6 & a_9 \\ a_9 & a_6 & M & a_2 \\ a_6 & a_9 & a_2 & M \end{bmatrix}, \quad D : \begin{bmatrix} a_8 & a_{10} & a_4 & O \\ a_{10} & a_8 & O & a_4 \\ a_4 & O & a_8 & a_{10} \\ O & a_4 & a_{10} & a_8 \end{bmatrix}.$$

There are 12 points in the Jacobian of $C_7$ bounded by the modulus 1: the 6 points in the Jacobian of $C$ as above and 6 extra points, $(1:1:0)$ (an extra point at infinity), $(0:1:1)$, $(0:-1:1)$, $(1:1:1)$, $(1:-1:1)$ (4 rational points) and the point $P := (x-p,0,1)$.

One finds 70 points bounded by the modulus 2 or 3, 694 points bounded by the modulus 4 and so on.

## 3. The Genetic Code Revisited

Our theory of the genetic code takes its inspiration in the symmetries observed in the DNA double helix and the biological steps leading to the conversion of transfer RNA

(tRNA) into the amino acids that code for proteins. Although 5-fold symmetry is inherent to the DNA double helix, being present in all its constituents, as shown in Figure 1a, the way to transcription into proteins needs an extra step in the spliceosome. The spliceosome is found within the nucleus of eukaryotic cells. Its role is to remove introns from the primary form of messenger RNA (mRNA) leaving the exons to be processed afterwards. This cutting process is called splicing. There is a heptamer ring, called the Lsm 1–7 complex, displaying a 7-fold symmetry in its protein constituents, as shown in Figure 1c [7]. Accounting for this observation, it is tempting to generalize our theory of the genetic code based on the 22 irreducible characters of the group $G_5 := (240, 105) \cong \mathbb{Z}_5 \rtimes 2O$, displaying the 5-fold symmetry, to the 29 irreducible characters of the group $G_7 := (336, 118) \cong \mathbb{Z}_7 \rtimes 2O$, displaying the 7-fold symmetry. The group $G_6$ is not an appropriate candidate for modeling the degeneracies of amino acids in the genetic code since nonirreducible character of $G_6$ is informationally complete, as shown in Table 1.

In Table A1 of the Appendix A, we reproduce the structure of the character table of the group $G_5$ and the assignments of its conjugacy classes to the proteinogenic amino acids as given in Ref. [1]. One drawback of the model is that there are only 2 sextets in the table while 3 of them are needed to fit the 3 sextets of the genetic code. In Table 4, this problem is solved since there are precisely three slots of degeneracy 6 in the character table of $G_7$. Table 4 shows entries proportional to the cosines of angles involved in the characters as $z_1 = 2\cos(2\pi/7)$, $z_2 = 2z_1$, $z_3 = -6\cos(\pi/7)$, $z_4 = \sqrt{2}$ and $z_5 = 2\cos(2\pi/21)$. Let us first concentrate on the 11 classes of degeneracy 2 of the group $G$. The character table contains the angle $2\pi/8$ through the 2 entries with $z_4 = 2\cos(2\pi/8) = \sqrt{2}$ as well as the angles $2\pi/7$ and $2\pi/21$ through the entries with $z_1$ and $z_{1,5}$, respectively. We choose not to assign amino acids to the 2 conjugacy classes with degeneracy 2 and $2\pi/8$ angle. The entries containing $z_5$ correspond to the hyperelliptic curve $y^2 = f(x)$ in (2) and the related Kummer surface. Then, there are two conjugacy classes with degeneracy 1 (ignoring the class with a trivial character) and 3, as expected. There are 3 classes with degeneracy 6 with entries containing $z_3$ and thus the angle $2\pi/7$, as we would expect. Finally, there are 10 slots for quartets but only 5 slots with entries containing the entry $z_5$ related to the Kummer surface. The first 4 slots are assigned to the 4 (degeneracy 4) amino acids and a slot is left empty. This leaves the freedom to assign this slot to the 21st proteinogenic acid Sec and the 22nd amino acid Pyl (compare to [1] or Table A1).

Now comes a question. Is the Kummer surface an attribute of RNA packings or is the proposed theory just another musing about the biological reality? We are not aware of any experiment featuring the Kummer surface in the biological realm. The physical link between messenger RNA (mRNA) and the amino acid sequence of proteins is a transfer RNA (tRNA). Corresponding to the three bases of an mRNA codon is an anticodon. Each tRNA has a distinct anticodon triplet sequence that can form three complementary base pairs to one or more codons for an amino acid. Some anticodons pair with more than one codon due to so-called wobble base pairing [18–21]. Considering the secondary and tertiary structure of tRNA, as well as the fact that the third position in the codon is not strictly red by the anticodon according to Watson-Crick pairing rules, Crick hypothesized that codon translation into a proteins is mainly due to the first two positions of the codon [18]. There are 16 groups of codons specified by the first two codonic positions and the level of degeneracy can be determined by them according to Lagerkvist's rules [19,20]. Our bet is that the 16 groups of codons correspond to the 16 singularities (double points) of the Kummer surface.

In the next section, we discuss the relevance of the Kummer surface in the context of 4-dimensional (space-time) quantum gravity.

**Table 4.** For the group $G_7 := (336, 118) \cong \mathbb{Z}_7 \rtimes 2O$, the table provides the dimension of the representation, the rank of the Gram matrix obtained under the action of the 29 -dimensional Pauli group, the order of a group element in the class, the angles involved in the character and a good assignment to an amino acid according to its polar requirement value. Bold characters are for faithful representations. All characters are informationally complete except for the trivial character and the one assigned to 'Met'. The entries involved in the characters are $z_1 = 2\cos(2\pi/7)$, $z_2 = 2z_1$, $z_3 = -6\cos(\pi/7)$, $z_4 = \sqrt{2}$ and $z_5 = 2\cos(2\pi/21)$ featuring the angles $2\pi/8$ (in $z_4$), $2\pi/7$ and $2\pi/21$.

| (336,118) $\mathbb{Z}_7 \rtimes (\mathbb{Z}_2 \cdot S_4)$ $\cong \mathbb{Z}_7 \rtimes 2O$ | | 1 | 1 | 1 | 2 | 2 | 2 | 2 | 2 | 2 | 2 |
|---|---|---|---|---|---|---|---|---|---|---|---|
| | dimension | 1 | 1 | 1 | 2 | 2 | 2 | 2 | 2 | 2 | 2 |
| | d-dit, d = 29 | 29 | 785 | $d^2$ | $d^2$ | $d^2$ | $d^2$ | $d^2$ | $d^2$ | $d^2$ | $d^2$ |
| | amino acid | . | Met | Trp | Cys | Phe | Tyr | . | . | His | Gln |
| | order | 1 | 2 | 3 | 4 | 4 | 6 | 7 | 7 | 7 | 8 |
| | char | Cte | Cte | Cte | $z_1$ | $z_1$ | $z_1$ | $z_4$ | $z_4$ | $z_{1,5}$ | $z_{1,5}$ |
| | polar req. | . | 5.3 | 5.2 | 4.8 | 5.0 | 5.4 | . | . | 8.4 | 8.6 |
| (336,118) | dimension | 2 | 2 | 2 | 2 | 3 | 3 | 4 | **4** | **4** | **4** |
| | d-dit, d = 29 | $d^2$ | $d^2$ | $d^2$ | $d^2$ | $d^2$ | $d^2$ | $d^2$ | $d^2$ | $d^2$ | $d^2$ |
| | amino acid | Asn | Lys | Glu | Asp | Ile | Stop | . | . | . | . |
| | order | 14 | 14 | 14 | 21 | 21 | 21 | 21 | 21 | 21 | 21 |
| | char | $z_{1,5}$ | $z_{1,5}$ | $z_{1,5}$ | $z_{1,5}$ | Cte | Cte | Cte | $z_{1,2}$ | $z_{1,2}$ | $z_{1,2}$ |
| | polar req. | 10.0 | 10.1 | 12.5 | 13.0 | 10 | 15 | . | . | . | . |
| (336,118) | dimension | **4** | **4** | **4** | **4** | **4** | **4** | 6 | 6 | 6 | |
| | d-dit, d = 29 | $d^2$ | $d^2$ | $d^2$ | $d^2$ | $d^2$ | $d^2$ | $d^2$ | $d^2$ | $d^2$ | |
| | amino acid | Val | Pro | Thr | Ala | Gly | . | Leu | Ser | Arg | |
| | order | 28 | 28 | 28 | 42 | 42 | 42 | 42 | 42 | 42 | |
| | char | $z_{2,5}$ | $z_{2,5}$ | $z_{2,5}$ | $z_{2,5}$ | $z_{2,5}$ | $z_{2,5}$ | $z_{1,3}$ | $z_{1,3}$ | $z_{1,3}$ | |
| | polar req. | 5.6 | 6.6 | 6.6 | 7.0 | 7.9 | . | 4.9 | 7.5 | 9.1 | |

## 4. Kummer Surface and Quantum Gravity

The Kummer surface first made its appearance in the Fresnel wave equation for light in a biaxial crystal [22,23]. The four singularities corresponding to the two shells in the Fresnel surface for a biaxial crystal lead to internal conical refraction as predicted by Hamilton in 1832. It is also known that for a magnetoelectric biaxial crystal, the Fresnel surface may display 16 real singular points, the maximal number permitted for a linear material whose dispersion relation is quartic in the frequency and/or wave number [23]. Although Kummer surface is relevant to electromagnetism, Ref. [24] discusses how it may also rely on gravity.

A theory of quantum gravity needs to conciliate our view of space-time as described by the general relativity and our view of particles and fields as described by quantum mechanics or quantum field theory. How is the Kummer surface related to attempts of formulating a theory of quantum gravity?

If one follows the historical perspective, our derivation of the Kummer surface in Section 2 relies on the work of Felix Klein in 1870 [22,25–27]. He introduces a quadratic line complex as the intersection $X = \mathrm{Gr}(2,4) \cap W$ of the Grassmann quadric $\mathrm{Gr}(2,4)$ in the five-dimensional Plücker space with another quadratic hypersurface $W$. The set of lines in $X$ is parametrized by the Jacobian $\mathrm{Jac}(C)$ of a Riemann surface of genus 2 ramified along 6 points corresponding to 6 singular quadrics. See [28] for the relation to string dualities.

Presently, in the classification by algebraic geometry, the Kummer surface is an example of a $K_3$ surface built from the quotient of an abelian variety $A$ by the action from a point $a$ to its opposite $-a$, resulting in 16 singularities [29,30]. The minimal resolution is the Kummer surface. There are many constructions of a $K_3$ surface $Y$ and it is known that all of them are diffeomorphic to each other. A $K_3$ surface is a compact connected complex manifold of dimension 2 with trivial first Chern class $c_1(Y) = 0$ so that the second Chern class (which corresponds to the topological Euler characteristic) is $c_2(Y) = 24$. Another important topological invariant of topological spaces is that of a Betti number $b_k$. Roughly,

$b_0$ is the number of connected components, $b_1$ is the number of one-dimensional 'holes', $b_2$ is the number of two-dimensional 'voids', and so on. For a $K_3$ surface, one has $b_0=b_4=1$, $b_1=b_3=0$ and $b_2=22$. This defines $Y$ as the unique unimodular even quadratic lattice of signature $(3, 19)$ isomorphic to $E_8(-1)^{\oplus 2} \oplus U^{\oplus 3}$, where $U$ the integral hyperbolic plane and $E_8$ is the well-known $E_8$ lattice. It is also known that the elliptic genus of a $K_3$ surface has a decomposition in terms of the dimensions of irreducible representations of the largest Mathieu group $M_{24}$ [31], a concept named 'umbral moonshine'. See also [32] for another view of the latter topic.

In the forefront of differential geometry, there is a connection of $K_3$ surfaces to quantum gravity in the concept of a Kähler manifold (with a Kähler metric). Such a manifold possesses a complex structure, a Riemannian structure and a symplectic structure. A $K_3$ surface admits a Kähler 'Ricci-flat metric' although it is not known how to write it in an explicit way. It is worthwhile to mention that a $K_3$ surface appears in string theory with the concept of 'string duality'-how distinct string theories are related-, see Ref. [13,28]. Another work relating quantum gravity and $K_3$ surfaces is in Ref. [33–35].

## 5. Discussion

Since the Kummer surface appears in our models of DNA/RNA packings of some protein complexes such as the hexamer and pentamer rings (the LSMs, MCMs and other biological complexes not given here) one can ask the question whether quantum gravity is relevant in such biological realms. It is a challenging question that we are not able to solve. Mathematics offers clues for models of nature. Biology is not a unified field as is quantum physics of elementary particles or the general relativity for the universe at large scales. We offered relationships between the n-fold symmetries ($n = 5$, 6 and 7) found in DNA and some proteins and the mathematics of Kummer surfaces. It is time to quote earlier work devoted to the possible relation between the microtubules of cytoskeleton and the field of quantum consciousness, e.g., [36]. The 13-fold symmetry is found in tubulin complexes [37]. Using the same approach than the one for DNA and hexamer/pentamer complexes, we can associate a finite group $G_{13} := (624, 134) \cong \mathbb{Z}_{13} \times 2O$ to such a 13-fold symmetric complex. Such a group possesses 50 conjugacy classes and the dimensions of representations are 1, to 2, 3, 4 and 6 as expected for this series of groups with factors $\mathbb{Z}_n$ and $2O$ in the semidirect product. It is found that a Kummer surface may be derived from some characters of $G_{13}$, as expected.

To conclude, one can observe a mathematical analogy between the way DNA/RNA organize and some theories of quantum cosmology based on string dualities. Lessons from one field may lead to progress in the other field. Finding similar algebraic structures in the theory of biology and quantum gravity may help to explain numerical coincidences as the ones observed in [38]. Should we talk about a new paradigm of 'biologic quantum cosmology' and revisit the philosophical foundations of quantum theory? A few papers are already written in this direction [39–41].

**Author Contributions:** Conceptualization, M.P., F.F. and K.I.; methodology, D.C., M.P. and R.A.; software, M.P.; validation, D.C., R.A., F.F. and M.M.A.; formal analysis, M.P. and M.M.A.; investigation, D.C., M.P., F.F. and M.M.A.; writing—original draft preparation, M.P.; writing—review and editing, M.P.; visualization, D.C., F.F. and R.A.; supervision, M.P. and K.I.; project administration, K.I.; funding acquisition, K.I. All authors have read and agreed to the published version of the manuscript.

**Funding:** Funding was obtained from Quantum Gravity Research in Los Angeles, CA.

**Data Availability Statement:** Not applicable.

**Conflicts of Interest:** The authors declare no conflict of interest.

## Appendix A

The table below was found in our paper [1]. An introduction to the DNA genetic code and the mention to some mathematical theories proposed before is discussed in this paper and not duplicated here.

**Table A1.** For the group $G_5 := (240, 105) \cong \mathbb{Z}_5 \rtimes 2O$, the table provides the dimension of the representation, the rank of the Gram matrix obtained under the action of the 22 -dimensional Pauli group, the order of a group element in the class, the entries involved in the character and a good assignment to an amino acid according to its polar requirement value. Bold characters are for faithful representations. There is an 'exception' for the assignment of the sextet 'Leu' that is assumed to occupy two 4-dimensional slots. All characters are informationally complete except for the ones assigned to 'Stop', 'Leu', 'Pyl' and 'Sec'. The notation in the entries is as follows: $z_1 = -(\sqrt{5}+1)/2$, $z_2 = \sqrt{5}-1$, $z_3 = 3(1+\sqrt{5})/2$, $z_4 = \sqrt{2}$, $z_5 = -2\cos(\pi/15)$, compare [Table 7] of [1].

| (240,105) $\mathbb{Z}_5 \rtimes (\mathbb{Z}_2 \cdot S_4)$ $\cong \mathbb{Z}_5 \rtimes 2O$ | | dimension d-dit, d = 22 amino acid | | | | | | | | | | |
|---|---|---|---|---|---|---|---|---|---|---|---|---|
| | dimension | 1 | 1 | 2 | 2 | 2 | 2 | 2 | 2 | 2 | 2 | 2 |
| | d-dit, d = 22 | $d^2$ | $d^2$ | $d^2$ | $d^2$ | $d^2$ | $d^2$ | $d^2$ | $d^2$ | $d^2$ | $d^2$ | $d^2$ |
| | amino acid | Met | Trp | Cys | Phe | Tyr | His | Gln | Asn | Lys | Glu | Asp |
| | order | 1 | 2 | 3 | 4 | 4 | 5 | 5 | 6 | 8 | 8 | 10 |
| | char | Cte | Cte | Cte | $z_1$ | $z_1$ | $z_4$ | $z_4$ | $z_{1,5}$ | $z_{1,5}$ | $z_{1,5}$ | $z_{1,5}$ |
| | polar req. | 5.3 | 5.2 | 4.8 | 5.0 | 5.4 | 8.4 | 8.6 | 10.0 | 10.1 | 12.5 | 13.0 |
| (240,105) | dimension | 3 | 3 | 4 | **4** | **4** | **4** | **4** | **4** | **4** | 6 | 6 |
| | d-dit, d = 22 | $d^2$ | 475 | 483 | 480 | $d^2$ | $d^2$ | $d^2$ | $d^2$ | $d^2$ | $d^2$ | $d^2$ |
| | amino acid | Ile | Stop | Leu,Pyl,Sec | Leu | Val | Pro | Thr | Ala | Gly | Ser | Arg |
| | order | 10 | 15 | 15 | 15 | 15 | 20 | 20 | 30 | 30 | 30 | 30 |
| | char | Cte | Cte | Cte | $z_{1,2}$ | $z_{1,2}$ | $z_{2,5}$ | $z_{2,5}$ | $z_{2,5}$ | $z_{2,5}$ | $z_{1,3}$ | $z_{1,3}$ |
| | polar req. | 4.9 | | | 4.9 | 5.6 | 6.6 | 6.6 | 7.0 | 7.9 | 7.5 | 9.1 |

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
