# Peer review of "Finite Groups for the Kummer Surface: The Genetic Code and a Quantum Gravity Analogy"

_quantumrep, doi:10.3390/quantum3010005_

Round 1

Reviewer 1 Report

This is a very demanding article that touches on the philosophical foundations of quantum theory. Therein the authors use again finite group theoretical tools to verify (revise / expand) own results recently published about a model for the genetic code of DNA.

This time, the group G7:=(336,118) was used to illuminate the special role of the heptameric ring complex Lsm1-7, which structure was analyzed by Zhou et al. in 2014. This complex is important in protein metabolism. However, seven-fold symmetry (pseudo-symmetry) is less frequently observed in nature. For instance, the little flower trientalis europaea (Siebenstern in German) has seven petals. The numbers 7, 11, 18 belong to the Lucas number series that play an important role in number theory besides the Fibonacci series. For instance, number 18 was found in connection with the alpha-helix, where 18 subunits perform 5 turns, and number 11 is found in the string theory of physics.

The authors used the hyperelliptic geometry of Kummer surfaces described by accompanied polynomials, and mention in passing Felix Klein's quartic, but it would be important to point to the duality of Klein's quartic as a quotient of the order-7 triangular tiling and the order-3 heptagonal one. Notice that the order 336 can be decomposed in 48 X 7 besides 112 X 3. A reference of the original work of Klein is missing: Klein, F. (1878) On the order-seven transformation of elliptic functions. Mathematische Annalen 14, 428-471.  Also the original work of the founder of the projective geometry Ernst Eduard Kummer should be quoted: Kummer, E. E. (1864) Über die Flächen vierten Grades mit 16 singulären Punkten. Monatsberichte der Königlichen Preußischen Akademie der Wissenschaften  zu Berlin, 246-260. The explicite Kummer surface expression K3 below sequence 173 on page 7, is the last term really 25x34?

At best, nature's evolutionary strategy used 'group theory' in a more statistical meaning, and group theory may only deliver an approximate construct of the truth.  One should be careful not to replace a construct by another better tuned construct by a further tuned one. Instead of proposing a group G13 for the important 13-fold symmetric complexes, why not varying the coefficients of a quartic? 

The concluding comments about revised understanding of quantum theory should point to the duality between matter and dark matter as piloting wavy entity of transformed moving matter and its consequences in biology.

The paper should be accepted for publication after some small proposed corrections.

Hermann Otto    

Author Response

The authors thank Referee 1 for his detailed report. We take good note of many useful suggestions for future work. We added the proposed references to Klein's work and the reference to foundational paper by Kummer. We also added a reference to a recent paper of the Referee that fits well our topic.

It is remarkable that the referee could point out a missing term in the mathematical expression of the Kummer surface on p. 7 line 173. Thanks again.

Reviewer 2 Report

The paper “Finite groups for the Kummer surface: the genetic code and a quantum gravity analogy” presents a theoretical view on DNA/RNA complexes, based on the generalization of their recent model of DNA/RNA genetic code. The authors observe a mathematical analogy between n-fold symmetries found in DNA molecule and some proteins and the mathematics of Kummer surfaces. I find the content very interesting, the work also reads smoothly, helpful illustrations have been added. The part of the work, regarding the elliptic and hyperelliptic curves, is review paper style. The proper references in the introduction have been listed. The work appears novel and interesting, it is deepening the knowledge of DNA in biology. The theory is documented with rigour and the results seem to be significant.
I consider it suitable for publication in Quantum Reports after minor revisions.

Some minor points on the paper which can improve the final readability:

1. In the introduction, it is not obvious how the authors relate the concepts of the models of quantum gravity theory and DNA/RNA complexes. What are the reasons for such analogies and have they confirmed by other authors?
2. How is it possible to use the presented theory in molecular biology? What are the consequences and other further application of proposed theory?

Author Response

We thank reviewer 2 for his report. We are glad that he found our work interesting and significant. To answer his comment 1, we put references [13,14] to M-theory in the introduction. We do not answer his comment 2 about more details about the usefullness of the presented theory to molecular biology. We leave it to the next paper to be submitted soon.